# The Determinants of the Growth of the European Bioplastics Sector—A Fuzzy Cognitive Maps Approach

Aikaterini Konti [1,*], Diomi Mamma [2,*], Nicolae Scarlat [1] and Dimitris Damigos [3]

1 Joint Research Centre (JRC), European Commission, 21027 Ispra, Italy; nicolae.scarlat@ec.europa.eu
2 Biotechnology Lab, School of Chemical Engineering, National Technical University of Athens (NTUA), 15780 Athens, Greece
3 Mining Engineering and Environmental Mining Lab, School of Mining and Metallurgical Engineering, National Technical University of Athens (NTUA), 15780 Athens, Greece; damigos@metal.ntua.gr
* Correspondence: aikaterini.konti@ec.europa.eu (A.K.); dmamma@chemeng.ntua.gr (D.M.)

**Abstract:** The extensive use of plastics and the environmental burden associated with their disposal have attracted the attention of scientists, politicians and citizens in Europe. In this frame, the EU has adopted a European Strategy for Plastics aiming, on the one hand, at reducing the use of plastic and, on the other hand, promoting their reuse in the context of a circular economy directly linked with the recently adopted Bioeconomy Strategy. Bioplastics could be an alternative to the conventional plastics, but they still have a limited share in the market. In this paper, Fuzzy Cognitive Maps (FCMs), a soft computing technique for analysing complex decision-making problems, is applied to identify the factors acting as drivers or barriers towards a bio-based plastics industry, their relative importance and the interactions between them. Experts with diverse backgrounds (technical experts, policy makers, industry executives) were interviewed in order to capture their perceptions and create a collective FCM capturing the strong and the weak points of the system. The collective FCM has a total number of 38 factors, which reflect the different approaches and knowledge of the experts. Overall, the "bio-based plastics" system is influenced mainly by the following factors: "EU Legislation", "Monomers purity", "Properties of the product", "Recycling potential", "Research & Development", "National Legislation" and "Production cost". The effect of the most significant political, social and techno-economic factors on the potential growth of the bioplastics sector has also been examined via simulations. The analysis demonstrated that the model is affected more (is more sensitive) to shifts in technoeconomic factors.

**Keywords:** bio-based plastics; plastics industry; FCMs; expert elicitation

## 1. Introduction

Plastics are everywhere in our everyday life; packaging, toys, vehicle parts, medical equipment, etc. are all made of plastic. The global production of plastics has increased twentyfold since the 1960s, reaching 322 million tonnes in 2015, and it is expected to double over the next 20 years [1]. In Europe, the plastics industry is a robust industry; 1.5 million people are employed in the sector which has an annual turnover of €350 billion, producing 18% of global plastics per year [2].

However, despite their wide use, plastics remain a problematic material essentially because of the pollution that their end of life is associated with. Plastics are among the items most commonly found on beaches, and they account for 50% of marine litter [3]. It has been estimated that 275 million tonnes of plastic waste was generated in 192 coastal countries, in 2010, with 4.8 to 12.7 million tonnes entering the oceans. Without waste management infrastructure improvements, the cumulative quantity of plastic waste available to enter the ocean from land is predicted to increase by an order of magnitude by 2025 [4]. Apart from the obvious ecological problems plastic pollution causes, it is harmful for local economies with fisheries and tourism being the main sectors affected. In exclusively

economic terms, the damage to marine environments has been estimated at 8 billion euros per year globally [1].

The European Union, taking into account the environmental burden the use of plastics implies, has recently adopted the "European Strategy for Plastics in a Circular Economy" [1]. According to the strategy, all plastic packaging should be recyclable by 2030. For the time being, less than 30% of such waste is collected for recycling, whereas landfilling and incineration rates remain high; 31% and 39%, respectively. As a result, it is estimated that 95% of the value of plastic packaging material is lost after a very short first-use cycle [5].

A possible solution to the above-mentioned problems could be the development and promotion of more sustainable materials. At the moment, plastics produced by alternative types of feedstock (e.g., bio-based plastics or plastics produced from carbon dioxide or methane) have a rather small share in the market.

Bio-based plastics are at least partially derived from biomass. Examples of biomass used in bioplastics include corn, sugarcane or other forms of cellulose. Currently, bio-based plastics account for between 0.5 and 1% of EU annual plastic consumption [1]. Up to now, the production cost of the bio-based plastics has been identified as a crucial factor that prevents their wider use. However, the importance of the bio-based sectors for a sustainable development in the EU has been acknowledged with the recent adoption of the updated 'Bioeconomy Strategy' [6]. More precisely, the updated strategy proposes a three-tiered action plan to do the following:

- Strengthen and scale up the bio-based sectors and unlock investments and markets,
- Deploy local bioeconomies rapidly across the whole of Europe,
- Understand the ecological boundaries of the bioeconomy.

In line with the 'Bioeconomy Strategy' and the 'Plastics Strategy', the European Commission has initiated the project 'Comparative Life Cycle Assessment of alternative feedstocks for plastic production' (http://eplca.jrc.ec.europa.eu/plasticLCA.html, accessed on 9 March 2022), underlining the importance of the development of plastics made of sustainable feedstocks. However, the transition to a more sustainable production and use of plastics is not only a technical issue. It is influenced by the policy framework as well as the consumption habits and preferences of the society.

Several studies have been carried out in the last decade concerning bioplastics. Previous studies aimed, among others, to investigate the environmental or technical performance of bioplastics (e.g., [7–10]), examine market diffusion and related consumer preferences (e.g., [11–13]), provide a review on the social, environmental and economic assessment of bioplastics [14], and examine recent technological advances [15,16] or the impact of standards and policies on bio-based plastics industry [17]. Moreover, in a recent study, a system dynamics approach was implemented to create a long-term projection (until 2030) of market perspectives for biodegradable bio-based plastics at the global and the European level [18]. Although the study recognises that global annual demand for bioplastics is affected by macroeconomic, technological, regulatory and social (referring to awareness) factors, the stock and flow model is mainly based on macro- and microeconomic factors, such as market price for fossil-based plastics, market price for biodegradable plastics, price elasticity of demand for biodegradable plastics, income elasticity of demand for biodegradable plastics, global annual Gross Domestic Product (GDP), process costs for biodegradable and fossil-based plastics, feedstock costs for biodegradable and fossil-based plastics, etc.

According to the review of the relevant literature, there are studies that have investigated individual economic, technical, environmental and other factors of the bioplastics industry. However, no studies were identified that attempted to provide a comprehensive picture. Aiming to fill this gap, the present paper intends to identify the factors acting as drivers or barriers towards a bio-based plastics industry, their relative importance and the interactions between them. In order to achieve this, the Fuzzy Cognitive Map method is implemented, eliciting the opinions of experts (academics, market experts, policy makers) in the field of bioplastics. The experts' 'bio-based plastics' FCM is then used to explore

via simulations the dynamics of the system, i.e., possible outcomes of the final state, when technoeconomic, social and political factors are altered in a predetermined way.

The rest of the paper is structured as follows: Section 2 describes the methodological approach. Section 3 presents the results of the outcome of the expert elicitation process, i.e., the collective 'bio-based plastics' FCM, analyses the main characteristics of the collective FCM based on graph theory indices and explores the behaviour of the system for the three categories of factors mentioned above. The final section discusses the main findings and conclusions drawn from this study.

## 2. Methodological Approach

### 2.1. Introduction to Fuzzy Cognitive Maps

Cognitive maps were introduced by Robert Axelrod as a tool aiming at representing social scientific knowledge and modeling decision making in social and political systems [19]. However, in real-life situations, relations between concepts are rarely clear. The integration of fuzzy logic to cognitive maps has led to Fuzzy Cognitive Maps (FCMs) which include fuzziness [20].

FCMs are weighted digraphs which consist of nodes and arcs. Nodes represent the concepts or factors used to describe the behaviour of a system, while arcs represent the relationships between these concepts as perceived by the participants [19]. More rigorously, each interconnection between two concepts has a weight which reflects the strength of the causal links between them. This weighted arc shows how strong is the impact of one concept on another. The weights of the arcs can be negative or positive indicating in this way a negative or positive effect of the one concept on the other. A weight equal to zero shows no interconnection between the concepts [21]. Spreadsheets or tables are used to map FCMs into comparison adjacency matrices [E] for further computation [22].

The main advantages of FCMs that have led to their wide use are [23] as follows:

- they are easy to understand by the stakeholders,
- they are easy to teach (to all the participants),
- have a high level of integration (needed for the complex issues),
- are not costly or time-consuming,
- give a system description.

Due to the abovementioned characteristics, FCMs have attracted the interest of the researchers and are currently used in a wide range of fields. For instance, FCMs have been used for determining the factors that influence the development of sustainable waste biorefineries [24], for environmental decision-making with stakeholder involvement [21], for analysing stakeholders' views about complex social–ecological systems and defining state outcomes through scenario analysis [25] etc. The applications of FCMs also include the development of climate policies through stakeholder engagement processes [23], the identification of policy drivers and private initiatives that may discourage unsustainable consumer behaviours with respect to food wastage [24], the creation of ecological models using expert and local people's knowledge [26], the exploration of risks and protective factors for maternal health in indigenous communities [27], the modeling of factors related to large-scale bioethanol production from biowaste [28] etc. Due to this growing interest, more reliable models that can better represent real situations and better analytical tools and indices (which are analysed below) have been developed.

### 2.2. FCMs' Structural Analysis

The matrix representation of FCMs can provide information on the structural properties of FCMs on the basis of Graph Theory and Networks analysis. FCMs can be analysed in relation to the number of concepts, connections, connection-to-concept ratio, and density [21,29]. The basic indices used for the analysis of the FCMs and the consequent comparisons between the FCMs of different persons and groups are described hereinafter.

The number of concepts refers to the number of components included in the model [26]. A higher number of connections indicates a higher degree of interaction between components in a model [26].

Transmitter variables are the components which affect other system components but are not affected by others [30]. On the contrary, receiver variables are the components which have only receiving functions; they are affected by other system components but have no effect [30]. Ordinary variables are the most common variables which have both transmitting and receiving functions; they influence and are influenced by other concepts [30].

Centrality score of individual variables reflects the relative importance of a system component to system operation. Centrality ($c_i$) is the most important measure for map complexity, coming from social network analysis. It is a measure of how connected the variable is to other variables and what the strength of this connection is. It is defined as the summation of variable's indegree ($id(v_i)$) and outdegree ($od(v_i)$) [20,21]:

$$c_i = od(v_i) + id(v_i) \tag{1}$$

The complexity index is the ratio of receiver to transmitter variables. It is a measure of the degree to which outcomes of driving forces are considered. Higher complexity indicates more complex systems thinking because more utility outcomes and implications and less controlling forcing functions are included in the system [26,30].

Hierarchy index ($h$) indicates the degree of 'democratic' thinking [31] and may indicate whether individuals view the structure of a system as top-down or whether influence is distributed evenly across the components in a more democratic nature. The hierarchy index is calculated using the following equation [26], where $N$ is the total number of factors and $od(v_i)$ is the outdegree:

$$h = \frac{12}{(N-1)N(N+1)} \sum_i \left[ \frac{od(v_i) - (\sum od(v_i))}{N} \right]^2 \tag{2}$$

When $h$ is equal to 1 then the map is fully hierarchical and when $h$ is equal to 0, the system is fully democratic. Democratic maps are considered much more adaptable to changes because of their high level of integration and dependence 26].

The density is an index of connectivity. It is calculated dividing the number of connections by the number of all possible connections. The higher the density, the more potential management polices exist [26,32].

*2.3. Construction of Collective FCMs*

The indices mentioned above are used in order to analyse the structure of the FCMs. It should be noted, though, that human understanding and expertise are the drivers for the construction of FCMs [26]. The input from the participants determines the design of an FCM. The process actually extracts the knowledge from the participants in order to investigate the problem's model and behaviour. The procedure according which the experts identify the concepts and the causal relationships among them and consequently assign weights to the interconnections between the concepts has been described in previous studies [19,21]. Given that an individual map reflects the subjective perspective/understanding of a person, it is not enough to present the complete description of the system under examination. Therefore, a higher number of participants is needed in order to capture all the important elements of the system. There is no fixed number of participants needed for the construction of a collective map, but accumulation curves correlating the number of concepts with the number of interviews are used in order to determine the sufficient representation of different perspectives [26].

Different aggregation techniques have been proposed in the literature in order to generate collective maps form the individual ones [21,26,27,33,34]. In the present study, similar concepts that appear in individual mental models have been condensed, and then a group model has been produced.

*2.4. Dynamic Analysis*

The dynamic of a FCM can be simulated analytically through a specific inference process. The construction of the FCMs and the adjacency matrix leads to the prediction of the steady state of the system. Mathematically, this steady state is represented by the following equation:

$$A_i^{(k+1)} = f\left( A_i^{(k)} + \sum_{\substack{j \neq i \\ j = 1}}^{N} A_j^{(k)} * W_{ji} \right) \qquad (3)$$

where $A_i^{(k+1)}$ is the value of concept *Ci* at simulation step *k+1*, $A_i^{(k)}$ is the value of concept *Cj* at step *k*, $W_{ji}$ is the weight of the interconnection between concept *Cj* and concept *Ci* and *f* denotes the transfer function used (e.g., logistic or sigmoidal) which gives values of concepts in the range [0, 1] [25–27].

FCMs also give the possibility to make hypothetical scenarios and explore how the system responds in the case of different conditions or different policy measures [26]. This scenario analysis is achieved using different input vectors (with activation levels between 0 and 1). In the case of *n* number of concepts, the input vector is 1 by *n*, the FCM adjacency matrix is $n \times n$, and the output is 1 by *n* [21].

In addition to understanding the structure and function of a system, the modeling process itself, i.e., developing an FCM with stakeholders, has also helped policy makers frame regulations in a way that takes into account the needs of stakeholders [25,26].

**3. Results**

*3.1. Survey Design*

In order to explore the potential development of the bioplastics sector using FCMs, nine experts were selected and interviewed. The interviews with the experts took place mostly on line between April and July 2021. In order to create a more realistic and complete model, opinions of stakeholders with diverse backgrounds were included taking into account the different approaches and interests. As a consequence, the experts participating in the present study come from different fields: academics/researchers in biotechnology/plastics, policy makers, and industry executives. Initially, the individual maps of the experts were constructed during the interviews. Then, the final collective FCM was drawn according to the procedure which has been described by Kontogianni et al. [29]. A software to create FCMs called Mental Modeler, available free of charge at http://www.mentalmodeler.org/ (accessed on 9 March 2022), was used for the needs of the study.

*3.2. Static Analysis of the FCMs*

Table A1, in Appendix A, lists (in alphabetical order) all the concepts mentioned by the interviewed experts. In total, 88 concepts which influence the uptake of the bioplastics sector were identified by the experts although some of them are practically the same with a different verbal description. A variety of factors ranging from legislative/political to technological and social ones have been stated by the experts. Some issues (for example legislation) are common and have been mentioned by almost all the participants in the survey, whereas others are more specific and are associated with the background of the interviewed experts. For instance, experts working in the field of polymerization and plastic production consider the purity of the monomers as an important factor influencing the overall production with significant impact on the production cost. On the other hand, experts coming from the policy sector focus more on the legislation and the public acceptance, neglecting the technical factors.

Table 1 analyses the individual FCMs with the help of graph theory.

**Table 1.** Graph theory indices for the individual FCMs.

| Expert | Total Components | Total Connections | Density | Connections per Component | Number of Driver Components | Number of Receiver Components | Number of Ordinary Components | Complexity Score | Hierarchy Index |
|---|---|---|---|---|---|---|---|---|---|
| 1 | 12 | 32 | 0.242 | 2.7 | 1 | 1 | 10 | 1 | 0.026 |
| 2 | 15 | 46 | 0.219 | 3.1 | 2 | 1 | 12 | 0.5 | 0.027 |
| 3 | 14 | 35 | 0.192 | 2.5 | 3 | 1 | 10 | 0.333 | 0.138 |
| 4 | 12 | 29 | 0.220 | 2.4 | 1 | 1 | 10 | 1 | 0.018 |
| 5 | 8 | 12 | 0.214 | 1.5 | 2 | 2 | 4 | 1 | 0.007 |
| 6 | 17 | 48 | 0.176 | 2.8 | 0 | 1 | 16 | - | 0.000 |
| 7 | 11 | 30 | 0.273 | 2.7 | 1 | 1 | 9 | 1 | 0.081 |
| 8 | 13 | 19 | 0.122 | 1.5 | 6 | 1 | 6 | 0.167 | 0.002 |
| 9 | 16 | 28 | 0.117 | 1.8 | 7 | 1 | 8 | 0.143 | 0.015 |
| Average | 13.1 | 31 | 0.197 | 2.3 | 2.6 | 1.1 | 9.4 | 0.6428 | 0.035 |

The number of components is, on average, 13.1 (ranging from 8 to 17), while the number of connections is 31, on average (minimum 12 and maximum 48) and the average density of the maps is 0.197. The average number of transmitter and ordinary (bidirectional) variables is 2.6 (with a range of 0 to 7) and 9.4 (from 4 to 16), respectively, and the number of receiver variables is 1.1. Concerning receiver variables, it should be noted that only one expert identified a second variable apart from the bioplastics sector. Regarding the connections per component, which is an index of the density between the components and the casual relation, they ranged from 1.5 to 3.1 (average 2.3). Finally, the average hierarchy index was 0.035 (between 0.000 and 0.138). As explained above, the influence is distributed across the different variables in a more democratic way when the hierarchy index is closer to 0.

For the construction of the collective FCM of the experts interviewed, some variables (expressing either the same concept in different words or similar concepts) were clustered. The aim of this process is to create less-complex maps that can serve as the baseline of different scenarios. The collective FCM of the present study is presented in Figure 1. The figure illustrates the complexity of the system and the multiple interactions between the different variables which are listed (in alphabetical order) in Table A2 in Appendix A.

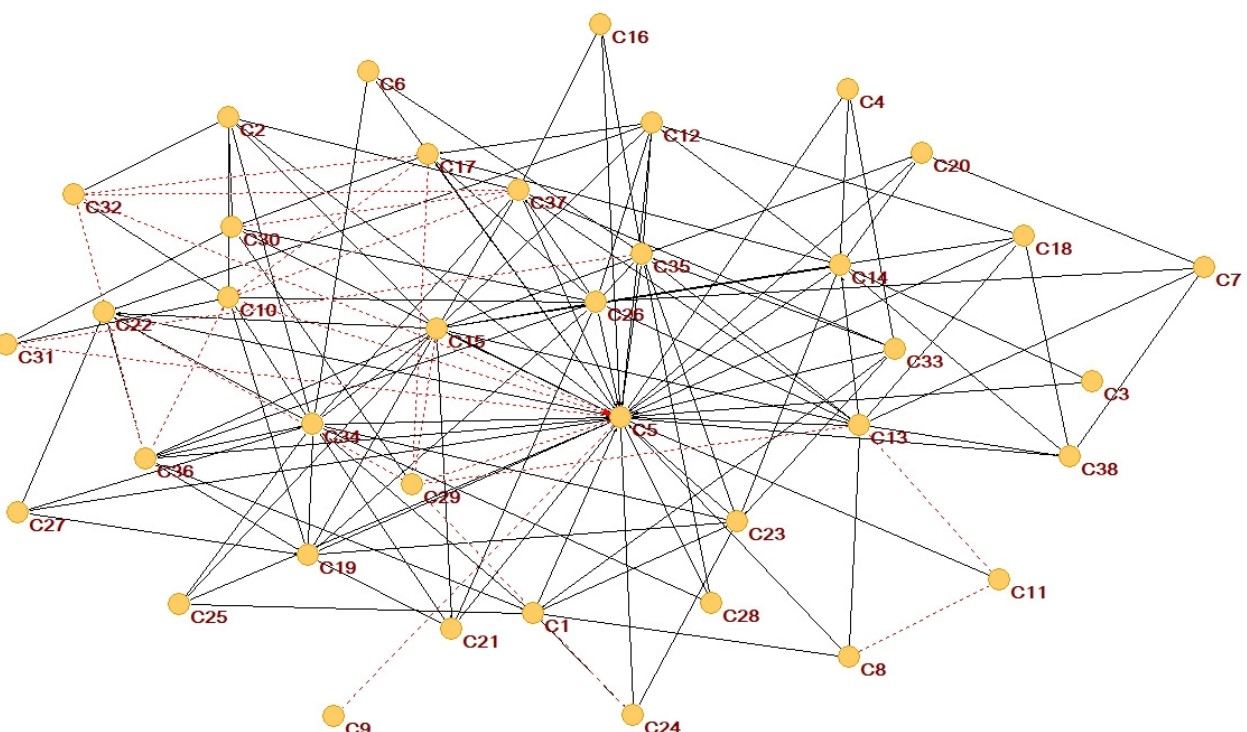

**Figure 1.** Collective FCM—the description of the factors is provided in Table A2, Appendix A (Source: created by the authors using the Pajek software—http://mrvar.fdv.uni-lj.si/pajek, accessed on 9 March 2022.)

Furthermore, Table A2 shows the clustering of the different concepts mentioned by the experts. For example, the variable 'Properties of the product' of the collective FCM incorporates the following components of the individual FCMs: "Comparability to conventional plastics", "properties", "properties of the new product", "similar properties to conventional products", "long term performance". It is noteworthy that clustering of the different components is a semi-subjective procedure that requires further development and standardization [35]. The collective FCM has a total number of 38 components. It should be highlighted here that a collective FCM is not the average of the individual FCMs but a different representation of the system characterized by totally new indices. This becomes clear from the graph theory indices of the collective FCM, which are presented in Table 2.

**Table 2.** Graph theory indices for the collective FCM.

| Total Components | Total Connections | Density | Connections per Component | Number of Driver Components | Number of Receiver Components | Number of Ordinary Components | Complexity Score | Hierarchy Index |
|---|---|---|---|---|---|---|---|---|
| 38 | 155 | 0.110 | 4.079 | 4 | 0 | 34 | 0 | 0.001 |

The collective FCM has 155 connections with 4.1 connections per component and density equal to 0.110. The most important difference between the collective and the individual FCMs regards, as expected, the hierarchy index. In the case of the collective FCM, the hierarchy index is 0.001, lower than the hierarchy index of each individual map as well as the average hierarchy index. The collective FCM has more components, which reflect the different approaches of the experts and the knowledge in different fields, resulting in a more complete description of the system. Therefore, the system is more democratic and more stable, i.e., the changes of individual components do not have such a big impact on the system. The most central concepts influencing the growth of the bioplastics sector with the corresponding indegree, outdegree and centrality are presented in Table 3. These concepts can be classified into three groups: political, social and technoeconomic. Overall, the most central variable is the "bioplastics sector" with a centrality of 7.12. "EU Legislation", "Monomers purity", "Properties of the product", "Recycling potential", "Research & Development", "National Legislation" and "Production cost" are the most 'central' concepts, meaning that they have the higher influence on the system.

**Table 3.** The most central concepts in the collective FCM.

| Component | Indegree | Outdegree | Centrality |
|---|---|---|---|
| Bioplastics sector | 7.10 | 0.02 | 7.12 |
| EU Legislation | 0.42 | 1.74 | 2.16 |
| Monomers purity | 0.11 | 1.84 | 1.96 |
| Properties of the product | 0.92 | 0.66 | 1.57 |
| Recycling potential | 0.16 | 1.40 | 1.56 |
| Research & Development | 0.49 | 0.93 | 1.43 |
| National Legislation | 0.98 | 0.30 | 1.28 |
| Production cost | 0.46 | 0.73 | 1.19 |
| Environmental awareness | 0.75 | 0.43 | 1.18 |
| Eco-friendly | 0.49 | 0.68 | 1.17 |

*3.3. Dynamic Analysis of the Collective FCM*

In order to explore the dynamics of the system (i.e., the interactions between the variables of the collective map), a number of simulations were carried out. The dynamic analysis can either focus on the equilibrium end-states or the transient behaviour during the iteration steps [25]. Using Kosko's "clamping methodology" according to which key variables are iteratively increased or decreased, a final vector of the procedure is generated and compared to the vector of the steady state [36]. Whereas the absolute values of the final vector are not of particular interest, the relative shifts in comparison to the steady state may be useful for policy makers indicating the potential dynamics of the system. The steady state of the system was predicted according to the procedure described in the section "Fuzzy Cognitive Map approach". More precisely, simulations were made by multiplying the initial state vector by the adjacency matrix of the aggregate FCM. Finally, simulations were conducted in order to explore the response of the system modifying the initial values of the concepts under study sequentially from 0.1 to 1. This range of values covers practically all the possible conditions (varying from no presence to the maximum

level). The software used for this analysis was the "FCM Tool" which works in Matlab environment [37].

The variables of the collective FCM were classified into three categories—technoeconomic factors, political factors and social factors—and the most central concepts (centrality > 1) of each category were selected for the analysis. The simulation process was carried out for each group independently and jointly for the three groups. Given that the current status of the parameters of interest (namely political, social, technoeconomic) is unknown, the predicted steady state has been used as the base case scenario against which the shifts of the individual variables have been quantified.

Simulations for a worst- and best-case scenario (with initial values of all the factors studied set to 0.1 and to 1, respectively) were carried out. Figure 2 summarizes the results as a difference between the two scenarios. The variables that were affected more by the shift of the parameters examined were 'Industrial production', 'Investment opportunities', 'Incentives for production' and the 'Bioplastics sector'. More precisely, a difference of 11.1% is expected for the 'Industrial Production' in the case of a shift from the worst to the best-case scenario. The difference in 'Investment opportunities' and 'Incentives for production' reaches 10.6% and 10.1%, respectively. Finally, the difference in the growth of the 'Bioplastics sector' between the two scenarios is 7.8%.

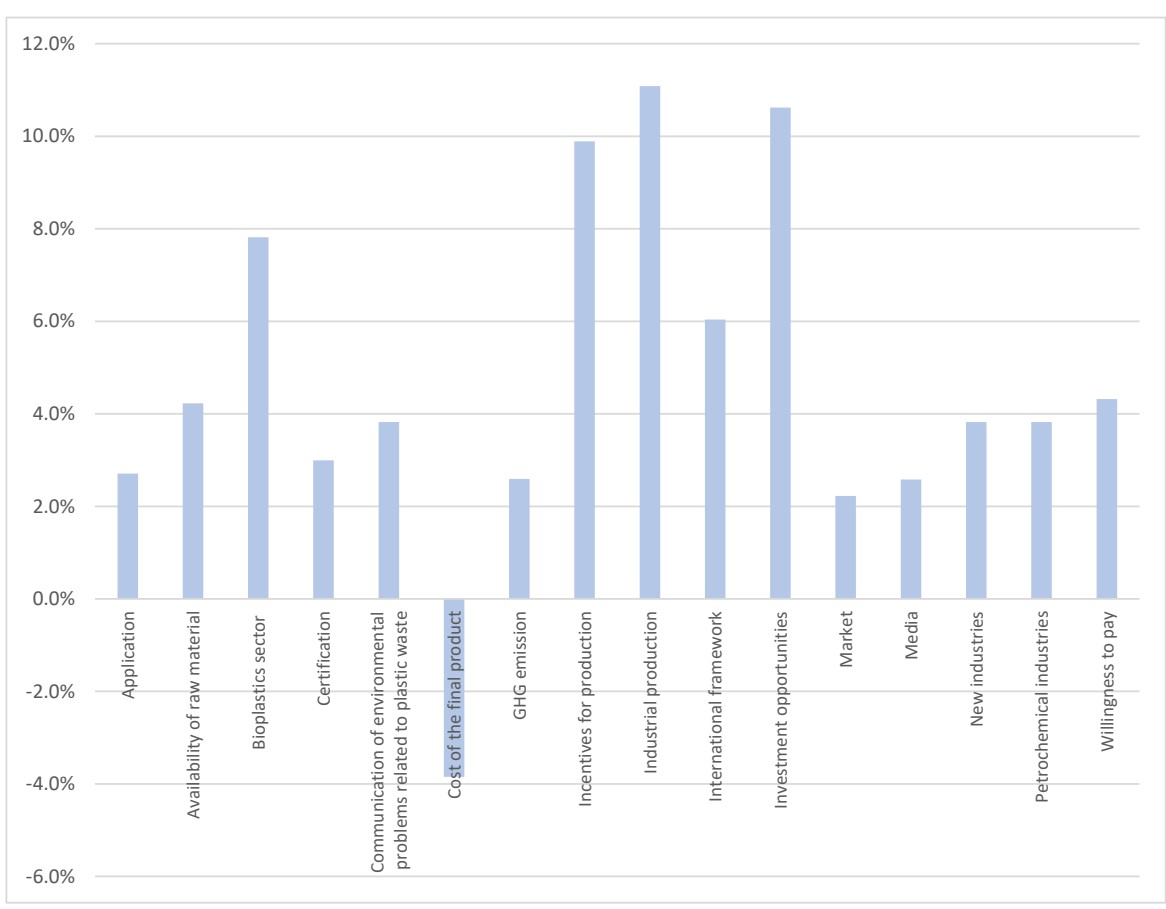

**Figure 2.** Comparison between the best and the worst-case scenario for all the variables examined.

The sensitivity analysis demonstrated that the model is affected more (is more sensitive) to shifts in technoeconomic factors. More specifically, "monomers purity", "properties of the product", "recycling potential", "research and development" and "production cost" are the variables of the FCM classified as technoeconomic factors. The results of the simulations are summarized in Figure 3. When the initial values of the chosen concepts are set to 0.1, the "Industrial production" drops by 5.2% whereas the "National legislation" and the "Investment opportunities" drop by 4.2% and 3.9%, respectively. A smaller decline

is observed in the "public acceptance", "incentives for production", "EU legislation" and "environmental awareness" ranging from 3.8 to 2.4%. On the other hand, when the technoeconomic factors take the highest value (i.e., 1) the "Industrial production" increases by 2.6% and the "Investment opportunities" by 2.0% compared to the baseline. The differences observed in the other variables are less than 2%.

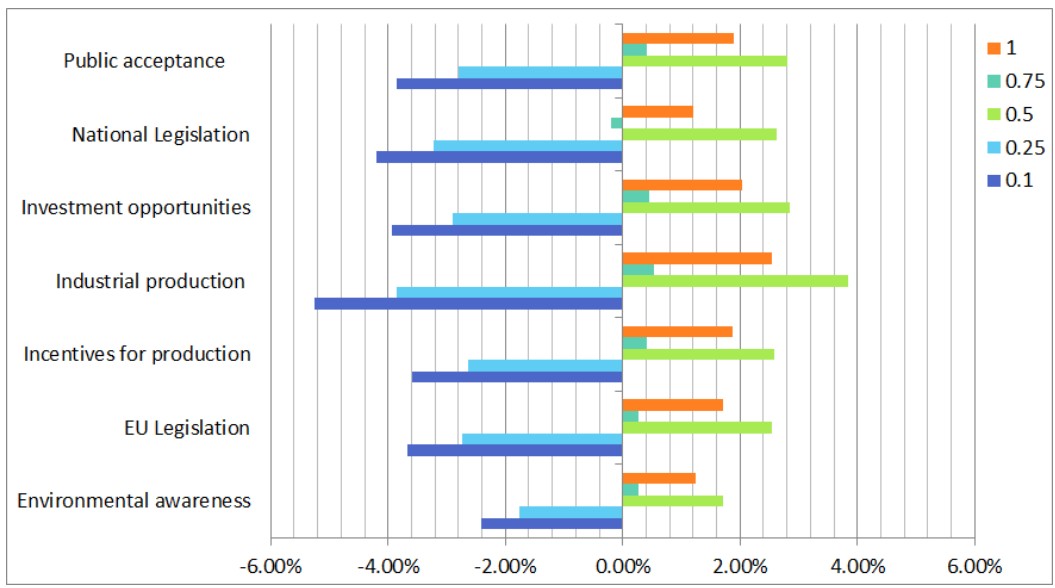

**Figure 3.** The effect of technoeconomic shifts on the other concepts of the collective FCM.

Concerning the political factors examined, namely "EU legislation" and "National legislation", the variables affected the most by the changes in the abovementioned factors are "Research & Development" and "Eco-friendly". Deviations ranging from −3.6% to +1.7% and 3.2% to +1.5%, respectively, in comparison to the baseline are observed with the lower value corresponding to an initial stimulus of 0.1 and the maximum to 1 (Figure 4).

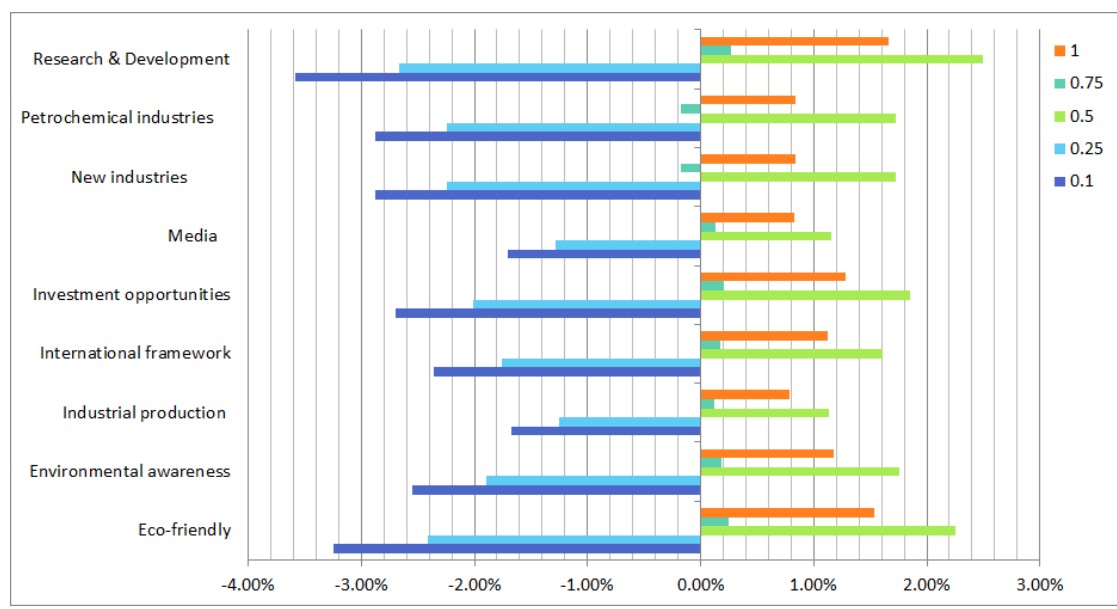

**Figure 4.** The effect of political shifts on the other concepts of the collective FCM.

At last, as far as the social variables (environmental awareness, eco-friendly, public acceptance, education) are concerned, less concepts are affected and the impact is lighter. A difference higher than ±2% in comparison to the baseline is observed in three variables,

namely "Willingness to pay", "National legislation", "Incentives for production". The deviations range from −3.0% to 1.2%, from −2.9% to 1.3% and from −2.7% to 1.1%, respectively, determined by the initial stimuli (Figure 5).

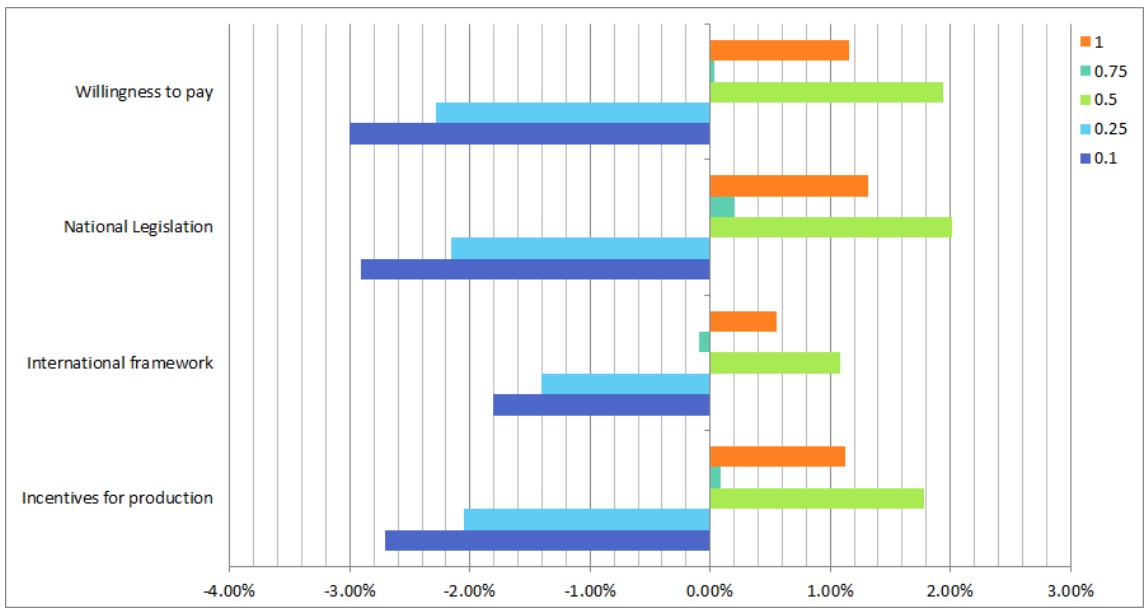

**Figure 5.** The effect of social shifts on the other concepts of the collective FCM.

## 4. Discussion and Conclusions

Large-scale bioplastics production can contribute to a reduction in the use of fossil-based plastics and the problems associated mainly with the use of fossil resources and the subsequent global warming. It can also pave the way to a sustainable bioeconomy which emerges as a priority in the EU. However, the large-scale production of bioplastics is a complex system comprised of many different factors, and its success is dependent on the collaboration of different actors such as policy makers, industry, scientists and citizens.

The present paper is an attempt to explore the bioplastics sector with the help of the FCMs approach. As far as we know, this is the first study which aims to model this system and therefore to identify its main components and their interactions. According to the analysis performed, the basic factors which mostly influence the system are "EU Legislation", "Monomers purity", "Properties of the product", "Recycling potential", "Research & Development", "National Legislation", "Production cost", "Environmental awareness", "Eco-friendly".

With regard to the dynamic analysis of the system, when the impacts of the shifts of the three different groups of variables (namely technoeconomic, political, social) were studied separately, it was found that the technoeconomic factors ("monomers purity", "properties of the product", "recycling potential", "research and development" and "production cost") had the strongest influence on the system. It is noteworthy that the direct impact of the technoeconomic factors on the central variable (the "bioplastics sector") is limited. However, the technoeconomic advances might lead to a significant increase in the production and can also trigger favourable legislative developments. A favourable policy framework is expected to result in more incentives and more R&D activities which will further support the growth of the sector.

It should be noted that the present study has some weaknesses, which should be kept in mind. The number of the experts interviewed for the construction of the individual FCMs and, finally, the collective FCM may be viewed as a limitation especially if the complexity of the system is taken into account. Moreover, the study has a European perspective, which does not necessarily reflect the situation in other parts of the world. In order to improve the model which describes the bioplastics sector, more reliable and representative maps

are needed. A better understanding of the system would be the prerequisite for better and more targeted policies. For example, a study with a wider geographical scope in which experts with different disciplinary backgrounds would participate would be a step towards this direction. Despite this limitation, this research points out the crucial parameters for the development of the bioplastics sector in EU and the ways they interact currently as well as in possible scenarios at which selected factors have been shifted. Therefore, it offers information that may prove useful to policymakers and relevant industries.

**Author Contributions:** Conceptualization, A.K. and D.M.; methodology, A.K and D.D.; software, D.D.; validation, A.K., D.M., N.S. and D.D.; investigation, A.K., D.M. and D.D.; writing—original draft preparation, A.K.; writing—review and editing, D.M., N.S. and D.D. All authors have read and agreed to the published version of the manuscript.

**Funding:** This research received no external funding.

**Conflicts of Interest:** The authors declare no conflict of interest.

**Disclaimer:** The views expressed here are purely those of the author and may not, under any circumstances, be regarded as an official position of the European Commission.

## Appendix A

**Table A1.** List of concepts declared by the experts during the construction of their individual maps.

| Components | Variable |
|---|---|
| C1 | Acceptance |
| C2 | Applications |
| C3 | Availability of feedstocks |
| C4 | Availability of raw materials |
| C5 | Availability of the new product |
| C6 | Awareness of the end user |
| C7 | Awareness of the society |
| C8 | Biodegradability |
| C9 | Bioplastics sector |
| C10 | Biotechnology |
| C11 | Certification |
| C12 | CO2 emissions |
| C13 | Communication of environmental problems related to plastic waste |
| C14 | Comparability to conventional plastics |
| C15 | Competitors-Conventional plastics industry |
| C16 | Consumption |
| C17 | Control of MW |
| C18 | Conventional industry |
| C19 | Cost |
| C20 | Cost of Production |
| C21 | Cost of the final product |
| C22 | Difficulties in management of plastic wastes |
| C23 | Eco-friendly |
| C24 | Economics |
| C25 | Education |

**Table A1.** *Cont.*

| Components | Variable |
|---|---|
| C26 | Education of the public |
| C27 | Environmental awareness |
| C28 | Environmental impact |
| C29 | Environmental Sustainability |
| C30 | Ethics |
| C31 | EU |
| C32 | EU legislation |
| C33 | EU Policy |
| C34 | European Union policy |
| C35 | Financial incentives for industry |
| C36 | GHG |
| C37 | Government |
| C38 | Government Policy |
| C39 | High Cost of Raw Material |
| C40 | Incentives for production |
| C41 | Income |
| C42 | Industrial Processes |
| C43 | Industrial production |
| C44 | Industrial technology |
| C45 | International framework |
| C46 | Investment opportunities |
| C47 | Legislation |
| C48 | Long term performance |
| C49 | Market |
| C50 | Marketing |
| C51 | Media |
| C52 | Monomers purity and quantity |
| C53 | National Legislation |
| C54 | New industries |
| C55 | NGOs |
| C56 | Old industry (fossil) |
| C57 | Petrochemical Industries |
| C58 | Policy framework |
| C59 | Political framework |
| C60 | Political Parties |
| C61 | Polymer Science and Technology |
| C62 | Price |
| C63 | Price of crude oil |
| C64 | Price of the new product |
| C65 | Priority of application depending on the product |
| C66 | Production cost |

**Table A1.** *Cont.*

| Components | Variable |
|---|---|
| C67 | Production technology |
| C68 | Productivity |
| C69 | Properties |
| C70 | Properties of the new product |
| C71 | Properties of the product |
| C72 | Public acceptance |
| C73 | Purity |
| C74 | R&D |
| C75 | Range of use |
| C76 | Raw Material |
| C77 | Recycle |
| C78 | Recycling potential |
| C79 | Reduction of environmental impact |
| C80 | Research & Development |
| C81 | Science and Technology |
| C82 | Seasonality raw material supply |
| C83 | Similar properties to conventional products |
| C84 | Society |
| C85 | Tailor-made products |
| C86 | Technological development for the production |
| C87 | Technology |
| C88 | Willingness to pay |

**Table A2.** Clustered variables.

| Components | Variable | Variables Clustered |
|---|---|---|
| C1 | Applications | Applications, Range of use, tailor-made products |
| C2 | Availability of raw materials | Availability of feedstocks, availability of raw materials, raw materials, seasonality of raw material supply |
| C3 | Availability of the new product | Availability of the new product |
| C4 | Biodegradability | Biodegradability |
| C5 | Bioplastics sector | Bioplastics sector |
| C6 | Certification | Certification |
| C7 | Communication of environmental problems related to plastic waste | Communication of environmental problems related to plastic waste |
| C8 | Consumption | Consumption |
| C9 | Control of MW | Control of MW |
| C10 | Cost of the final product | Cost, cost of the final product, price, price of the new product |
| C11 | Difficulties in management of plastic wastes | Difficulties in management of plastic wastes |
| C12 | Eco-friendly | Eco-friendly, environmental impact, environmental sustainability, reduction of environmental impact |

**Table A2.** *Cont.*

| Components | Variable | Variables Clustered |
|---|---|---|
| C13 | Education | Education, Education of the public |
| C14 | Environmental awareness | Awareness of the end user, awareness of the society, environmental awareness |
| C15 | EU Legislation | EU, EU legislation, EU policy, European Union policy |
| C16 | GHG emissions | CO2 emissions, GHG |
| C17 | Incentives for production | Financial incentives for industry, incentives for production |
| C18 | Income | Income |
| C19 | Industrial production | Industrial production |
| C20 | International framework | International framework |
| C21 | Investment opportunities | Investment opportunities |
| C22 | Market | Market |
| C23 | Marketing | Marketing |
| C24 | Media | Media |
| C25 | Monomers purity | Purity, Monomers purity and quantity |
| C26 | National Legislation | Government, government policy, legislation, national legislation, policy framework, political framework |
| C27 | New industries | New industries |
| C28 | NGOs | NGOs |
| C29 | Petrochemical industry | Competitors-Conventional plastics industry, Conventional Industry, old industry (fossil), petrochemical industries |
| C30 | Price of crude oil | Price of crude oil |
| C31 | Production cost | Cost of production, high cost of raw materials, production cost |
| C32 | Productivity | Productivity |
| C33 | Properties of the product | Comparability to conventional plastics, properties, properties of the new product, similar properties to conventional products, long term performance |
| C34 | Public acceptance | Acceptance, public acceptance, society |
| C35 | Recycling potential | Recycle, recycling potential |
| C36 | Research & Development | Biotechnology, polymer science and technology, R&D, research and development, science and technology |
| C37 | Technology | Industrial Processes, industrial technology, production technology, technological development for production, technology |
| C38 | Willingness to pay | Willingness to pay |

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
