# Peer review of "The Determinants of the Growth of the European Bioplastics Sector—A Fuzzy Cognitive Maps Approach"

_sustainability, doi:10.3390/su14106035_

Round 1
Reviewer 1 Report
The paper deals with the definition and application of a system dynamics model to understand the interplay between the fossil-based plastics and bioplastics.
The topic is in line with aim and scope of the journal.
Apparently, the authors do not introduce any significant methodological advance, but propose an interesting application of existing system dyanmics modelling framework feed by information provided by real stakeholders.
The work seems interesting, but it presents some flaws: the opinion of the reviewer is that the paper should be rejected and reconsidered after major revisions. By reading the paper, the overall impression is that the paper has written in a hurry, without putting appropriate care in setting the background and literature review, describing methodology, discussing the obtained results, and especially providing some sort of model validation.
Here are some specific comments:
- abstract: the context and methods are extensively descripted, while results are not well and not effectively summarized.
- keywords: genetive should not be applied to "system". moreover, system modelling, bioeconomy, renewable resources, are too generic for this context.
- L73-78: the objective of the work is not clearly explained. in particular, the methods and models you actually adopted for this paper seem a bit far from your initial claim "model the interplay between fossil- [...]". Moreover, in L76, the sentence "... explore the dynamics of the system" is confusing: if your objective is to define and use a system dynamics model, i suggest to state this more clearly, and to better specify what is the final purpose of the model. I suggest to better reshape the aim of the work.
- in the introduction, it is not so clear what your study is adding compared to other studies in the literature. A thorought but concise literature review section is missing in the introduction (while it is too extensive in the methodology section, just related to the application of the methods), hence it is difficult understanding the criticalities and gaps in current research that justifies the proposed research.
- analytical expressions and equations should be numbered. moreover, a symbols/nomenclature list is missing: it is very difficult to understand the meaning of the equations (e.g. L144: what are these symbols? ... moreover, please avoid multiple "=" signs in the same line).
- literature references are put in the paper with different style (e.g. L142, L148). please provide references in line with the paper guidelines. moreover, please avoid lumped references (e.g. [1], [2]), but provide a description to each reference.
- methodology section is very difficult to be understood. it seems written in a hurry, and without the appropriate care in defining the adopted models and equations. the paper should be more rigorous in defining the adopted approach. moreover,
- i suggest to put section 3.3. in appendix, because the tables and figures are so difficult to be understood. moreover, table 1 is incomplete.
- figure 1 is not understandable at all. in general, all the figures are badly formatted. and poorly commented.
- in general, the conclusions are poor and mostly "general", not very focused on data and results from the model.
Reviewer 2 Report
The reviewed manuscript “What will define the growth of the European bioplastics sector? A Fuzzy Cognitive Maps approach”, deals with the growth of the European bioplastics sector, and will be interesting for Sustainability readers, after revision.
The title: What will define the growth of the European bioplastics sector? A Fuzzy Cognitive Maps approach – “The determinants the growth of the European bioplastics sector A Fuzzy Cognitive Maps approach” – you could change the question in the title on sentence equivalent.
Aims: More precisely, the present work aims, on the one hand, to identify the factors, challenges, burdens towards a plastics industry based on bioplastics and, on the other hand, to explore the dynamics of the system. (l.74-75) - whether all objectives have been achieved, should be improved;
Discussions – the literature on the method and the state of the knowledge of this scientific problem should be enriched; one long sentence with many “for” is to simply for discussion;
Conclusion: lack of conclusions, you should give a few main points/ideas from the study.
What does “bio-based plastics” mean? – please, shortly define this notion;
Chapter 3.3. Figures, Tables, and Schemes - whether it is possible to include tables and graphs in the text and dispense with such a chapter;
(…) groups of variables (namely techno-economic, political, social) were studied separately, it was found that the techno-economic factors had the strongest influence on the system. (I. 366-367) – give some indices - techno-economic, political, social.
The procedure according which the experts identify the concepts (l. 166) - could you be more specific about the experts?’
Table 2 Clustered variables – perhaps move to Appendix too?
What is the novelty, or the research for the future?
Some specific comments:
l. 63 – 65: commas, dots at the end of sentences;
l. 68: Comparative Life Cycle Assessment of alternative 68 feedstock for plastic production’1 – it will be correct;
l. 73: …work… – the aim of the study – will be better;
l. 93: the concepts [9]. Spreadsheets (…) – with dote at the end of the sentence;
l. 99 – 100: - are not costly or time-consuming, and;
l. 99 – and, give a system description;
l. 102 – 118: For instance, FCMs have been used for determining the factors that influence the development of sustainable waste biorefineries [12], for environmental decision-making with stakeholder involvement [9], for analyzing stakeholders’ views about complex social-ecological systems, and defining state (…) communities [14], for (…) for developing climate policies through stakeholder engagement processes [26], etc. – you should modify “a bit” this fragment of the text;
l. 119: [26], etc.. – remove, one dote at the end of the sentence;
l. 152: space entry formula and text;
l. 177: [27], [31] ,[25] ,[9], [32] – it should be improve;
l. 326: give one-two sentence to introduce par 3.3, and some comments for another table, scheme, and so on; please, check table 1- invisible fragment;
l. 345: Figure 2: captions on the axes overlap;
l. 366: groups of variables (namely techno-economic, political, social) were studied separately, it was found that the techno-economic factors had the strongest influence on the system. – give the literature;
l. 397: what is the cognitive value of Appendix”? Is it useful in this manuscript?;
Reviewer 3 Report
1. In the Future Try to refer latest articles
2. In the future add optimization work
Reviewer 4 Report
The paper entitled “What will define the growth of the European bioplastics sector? A Fuzzy Cognitive Maps approach” by Konti et al. describes a first modeling of the European bio-based plastic sector by Fuzzy Cognitive Maps. In the work, several experts on the topic with different backgrounds have been interviewed and their main ideas have been related to each other and, in a mathematical way, “quantified”. This allows to find what of these ideas are the most important and, somehow, to consider different scenarios.
The manuscript is clear, academic, and well-explained. In my opinion, this paper is very, very interesting with results and conclusions that can be very useful for policymakers, researchers, industries, and, in general, citizens concerned about plastics. In addition, it can be the seed of other more ambitious works.
I do not have any comment or suggestion.
Congratulations to the authors.
Round 2
Reviewer 2 Report
no suggestions for authors, but in future papers should "practice" leading the discussion.
This manuscript is a resubmission of an earlier submission. The following is a list of the peer review reports and author responses from that submission.